# Molecular Targets and Promising Therapeutics of Triple-Negative Breast Cancer

**DOI:** 10.3390/ph14101008

**Published:** 2021-09-30

**Authors:** Won-Ji Ryu, Joo Hyuk Sohn

**Affiliations:** 1Avison Biomedical Research Center, Yonsei University College of Medicine, Seoul 03722, Korea; wjryu711@yuhs.ac; 2Division of Medical Oncology, Department of Internal Medicine, Yonsei University College of Medicine, Seoul 03722, Korea

**Keywords:** triple-negative breast cancer, heterogeneity, molecular target, new drugs

## Abstract

Triple-negative breast cancer (TNBC) is one of the most heterogeneous diseases in solid tumors and has limited therapeutic options. Due to the lack of appropriate targetable markers, the mainstay therapeutic strategy for patients with TNBC has been chemotherapy for the last several decades. Indeed, TNBC tumors have no expression of estrogen receptor, progesterone receptor, or human epidermal growth factor receptor 2 (HER2); therefore, they do not respond to hormone therapy and HER2-targeted therapy. In this review paper, the molecular heterogeneities, possible therapeutic targets, and recently approved and upcoming drugs for TNBC will be summarized.

## 1. Introduction

Breast cancer is a heterogenous disease. Hormone receptor-positive breast cancer, covering luminal A and luminal B subtypes (70–80%) in gene expression profiles, is quite dependent on the hormone receptor pathway, so its mainstream treatment is endocrine therapy with or without targeted agents to interrupt estrogen-signaling pathways and other critical pathways, such as cyclin-dependent kinase (CDK) 4/6 and phosphoinositide 3-kinase (PI3K)-AKT-mechanistic target of rapamycin (mTOR) pathway, for cancer survival. Human epidermal growth factor receptor 2 (HER2)-positive breast cancer comprises a HER2-enriched subtype and luminal B subtypes (10–15%) in molecular profiling and is addicted to the HER2 oncogene; thus, HER2-targeted therapies combined with cytotoxic chemotherapy are regarded as the standard treatment. By contrast, triple-negative breast cancer (TNBC) lacks these three receptors (estrogen receptor (ER), progesterone receptor (PR), and HER2) and, thus, is a diagnosis of exclusion showing the basal subtype in molecular profiling. In addition, due to advanced “multi-omics” technology, TNBC itself has been revealed to be a heterogeneous disease with several subtypes according to cancer biology [1].

Given that only 10–15% of all breast cancer cases are TNBC, further classification into smaller subgroups has not been clinically practical. Therefore, most of the clinical trials with targeted agents for TNBC included all TNBC subtypes, even after full awareness of it as a heterogeneous disease [2,3]. Basically, it would be difficult to obtain a meaningful result in these trials since a specifically targeting agent won’t work on such a mixed group of diseases. Recently, there have been several trials that limited enrollment to only participants with a particular subtype, who are most likely to respond to the targeted agent. One good example is poly (ADP-ribose) polymerase (PARP) inhibitors for TNBC patients with germline *BRCA1/2* mutations, accounting for approximately 10% of TNBC patients. [4,5,6]. In this regard, this review discusses molecular targets and recently proved and upcoming therapeutics of TNBC.

## 2. Molecular Subtypes of TNBC

With the development of “omics” (transcriptomic and genomic) technology, breast cancers were classified into the following five intrinsic subtypes (PAM50) by analyzing their gene expression pattern rather than the receptor expression status: basal-like, luminal A or B, HER2-enriched, or normal-like [2,3,7]. These pivotal studies found that breast cancers with triple-negative characteristics do not always overlap with the basal-like subtype, which means that although approximately 90% of TNBC cases are basal-like breast cancer (BLBC) [8], the rest of TNBC cases show the other breast cancer subtypes, such as HER2 enriched or luminal B subtypes.

Studying genetic and molecular heterogeneity of patients with TNBC, Lehmann and a group of colleagues [9] defined six TNBC subtypes according to gene expression clustering: basal-like 1 (BL1), basal-like 2 (BL2), mesenchymal (M), mesenchymal stem-like (MSL), immunomodulatory (IM), and luminal androgen receptor (LAR) (Table 1).

The BL1 subtype exhibits abnormal expression of genes related to cell cycle regulation and DNA repair (*MYC*, *PIK3CA*, *KRAS*, *FGFR1*, *AKT2*, *BRCA1*, *TP53*, and *RB1*) [9,36]. The BL2 subtype shows abnormal activation in various pathways related to cell growth (EGFR, MET, Wnt/b-catenin, mTOR, and IGF1R pathways) [9,36]. The IM subtype is marked by several activated pathways involved in immune cell signaling: CTLA4, NK cell, Th1/Th2, NF-kB, TNF, T cell signaling, dendritic cell, and BCR pathways. The histological phenotype of IM tumors is similar to medullary breast cancer [9,37]. In the M subtype, the cell migration or epithelial-mesenchymal transition (EMT)-related Wnt, TGFb, and extracellular matrix (ECM) pathways are highly activated. The MSL subtype exhibits a higher expression of stemness-related genes than the M subtype. The histological characteristics of these mesenchymal subtypes are sarcoma-like or squamous epithelial cell-like phenotypes, and they overlap with metaplastic breast cancer [9]. The LAR subtype does not express ER and PR, but hormone-related pathways are activated, such as androgen and estrogen metabolism, steroid biosynthesis, tyrosine metabolism, and ATP synthesis [9]. For the LAR subtype, androgen receptor (AR) expression is confirmed through immunohistochemistry. This subtype shows a gene expression pattern that is the unique from other TNBC subtypes and its histological type is similar to an apocrine tumor [11].

The TNBC subtypes were further compared with the intrinsic molecular (PAM50) subtypes to analyze their relevance [36]. Most TNBC subtypes were classified as basal-like subtypes in PAM50-based profiling (BL1, 99%; BL2, 95%; IM, 84%; and M, 97%), except for MSL (basal-like, 50%; normal-like, 28%; and luminal B, 14%) and LAR (HER2-enriched, 74%; and luminal B, 14%) subtypes. Furthermore, in order to discover a new subtype-specific target, recent studies have attempted to reanalyze TNBC molecular profiles for classification [38,39].

## 3. Potential Molecular Targets of TNBC

### 3.1. DNA Repair Pathway

BRCA1 and BRCA2 are considered to be tumor suppressor genes. They are critical for maintaining DNA replication error control, cell division, DNA repair, and apoptosis involving the homologous recombination repair pathway [17,18]. The germline mutation of *BRCA1* or *BRCA2* carriers ultimately develop breast cancer in 60–70% in their lifetime, and their mutation in TNBC occurs with a frequency of 10% [4,5,12]. Interestingly, tumors with the basal-like phenotype have been frequently observed harboring *BRCA1* mutations, while ER-positive breast cancers are closely correlated with *BRCA2* mutations [13,14].

*BRCA1* and *BRCA2* have a critical role in double-strand break repair via the homologous recombination pathway (Figure 1). Patients with the germline mutations of these genes show a loss of heterozygosity and pathologically high-grade tumors [40]. Tumors with these mutations are sensitive to DNA-damaging drugs and show a selective toxicity to the inhibitors of PARP responsible for single-strand break repair [6].

In addition to BRCA1/2, the DNA repair pathway is also regulated by many other proteins, such as ATM, PALB2, RAD51, BARD1, BRIP1, PARP1, TP53, and CHK2 [30,41]. Therefore, the homologous recombination deficiency (HRD) phenotype has been observed in tumors without mutations of *BRCA1* and *BRCA2*, which have been defined as “BRCAness” [4,5]. Some *BRCA*-proficient TNBC patients have shown this HRD phenotype and a high response to DNA-crosslinking agents [31,42].

### 3.2. PI3K/AKT Pathway

The *PIK3CA* mutation (10.2%) is the second major gene aberration, followed by the *TP53* mutation, found in patients with TNBC [4,5]. These hotspot mutations in the *PIK3CA* gene induce the abnormal activation of the PI3K pathway (Figure 1). *PTEN* loss (9.6%), a negative regulator of the PI3K pathway, has a mutually exclusive relationship with the *PIK3CA* mutation [5]. These alterations result in the activation of AKTs, which could be attractive targets for TNBC therapy.

In addition, PI3K plays a role of binding and stabilizing double-strand breaks via interaction with the homologous recombinant complex [43]. Therefore, the inhibition of PI3K results in homologous recombination impairment and then makes a state similar to a BRCA1/2-deficient tumor. Consistent with this observation, blocking PI3K could make TNBC tumors more sensitive to PARP inhibitors or DNA-damaging agents.

### 3.3. RAS/MAPK Pathway

Aberrant mutations in major factors of the RAS/MAPK pathways are rare in patients with TNBC [4,9,44]. However, gain-of-function mutations of *RAF*, *HRAS*, and *KRAS* have been found in some TNBC cell lines [45]. In addition, TNBC cell lines without these mutations have also been reported to show increased activation of the RAS/MAPK pathway (Figure 1).

The overexpression of epithelial growth factor receptor (EGFR) is known to be a common feature of BLBC and TNBC [32]. As an upstream molecule of the RAS/MAPK pathways, EGFR has an important role in signaling transduction. The gene copy number alteration of EGFR is highly increased in patients with TNBC [33]. In addition, various tyrosine kinase receptors, such as FGFR1, IGF1R, ERRB3, and ERBB4, are upregulated in TNBC [32].

In the RAS/mitogen-activated protein kinase (MAPK) pathway, one of the targetable proteins for patient with TNBC is MEK. TNBC and BLBC cell lines tend to be more sensitive to MEK inhibitors than other subtypes [46]. In addition, MEK is known to regulate the stability of c-Myc, which can be degraded after treatment with MEK inhibitor in TNBC [47,48]. However, the degradation of c-MYC via single agent MEK inhibition has an adverse effect that induces the activation of receptor tyrosine kinases (RTKs) [48]. Further studies are needed to overcome the resistance of single treatment with MEK by combining it with RTKs-targeted monoclonal antibodies.

Another cause of RAS/MAPK pathway activation is the loss of the dual specificity phosphatase 4 gene (*DUSP4)*, a negative regulator of extracellular signal-regulated kinase 1/2 (ERK1/2) and c-Jun N-terminal kinase 1/2 [49]. In particular, DUSP4 is known to be associated with the activation of the RAS/MAPK pathway in the BLBC subtype. In preclinical studies, it was found that resistance to chemotherapeutic agents is induced via the loss of its function, negatively regulating the RAS/MAPK pathway [49]. The loss of *DUSP4* enhances the maintenance of cancer stem cell populations, which can be abolished when treated with the inhibitors of the RAS-ERK pathway [50].

### 3.4. Androgen Receptor Pathway

The LAR subtype shows nine times higher mRNA expression of the AR than the other TNBC subtypes, predisposing LAR tumors to be susceptible to AR antagonists [9]. AR is expressed in 10–50% of patients with TNBC [51,52,53]. Patients with the LAR subtype tend to have a poor chemotherapy response and rarely achieve pathological complete response (pCR) in neoadjuvant chemotherapy [19,54,55].

AR is one of the steroid hormone receptors in the nuclear receptor family. In the absence of androgen, AR localizes to the cytosol, but after ligand binding, the receptor-hormone complex moves to the nucleus and increases the expression of the target genes (Figure 1) [56,57]. In addition, AR signaling is also activated through ERK-mediated signaling with PI3K, Src, and RAS [58,59]. LAR-subtype TNBC cell lines with increased AR expression were found to frequently carry *PIK3CA*-activating mutations, demonstrating a correlation between AR dependency and the PI3K pathway [60]. In addition to AR inhibitors, these cells are sensitive to PI3K inhibitors, as ER-positive breast cancer cells with *PIK3CA* mutations succumb to PI3K inhibitors such as alpelisib [61].

### 3.5. Other Pathways

Various studies are ongoing to identify candidates targeting metastatic TNBC. One of the targets is epigenetic DNA methylation. The methylation on *BRCA1* promoter is correlated with poorer overall survival (OS) and recurrence-free survival (RFS) in TNBC [62]. A recent clinical study reported that hypermethylation patterns can predict higher pCR rate after neoadjuvant chemotherapy in patients with TNBC [63]. Phase I/II studies of histone deacetylase inhibitors have been under investigation for patients with metastatic TNBC (NCT02393794).

Breast cancer stem cells have been considered as a primary cause of recurrence and metastasis due to the repopulating ability from a single cell [64]. Breast cancer stem cells have a slower growth rate than breast cancer non-stem cells, making them less responsive to chemotherapy [65]. In TNBC, BLBC subtype cells are known to be enriched with cancer stem cells. Preclinical and clinical studies have shown that Wnt/b-catenin [28], Notch [66], Hedgehog [67], JAK/STAT [68], and RAS/MAPK [50,69] pathways could enhance breast cancer stem cell populations.

## 4. Immune System in TNBC

The major cancer-related immune response is adaptive immunity, including cytotoxic CD8 T-lymphocyte in the immune microenvironment of cancer. The genomic abnormality of cancer causes the presentation of neo-antigenic peptide binding to major histocompatibility complex (MHC) on the surface of tumor cells. As a result, cytotoxic T-lymphocytes recognize the neo-antigenic peptide and lyse the tumor cells.

Breast cancer shows a high level of genomic instability and could stimulate cancer-related immune system. A representative phenomenon of immunogenic activation is lymphocytic infiltration in the tumor microenvironment. TNBC has a tendency to show a higher level of tumor-infiltrating lymphocytes (TILs) than the other breast cancer subtypes [10,70]. TNBC is subject to showing higher genomic instability and mutation burden, which stimulates the adaptive immune system through presenting a number of neo-antigens [71]. Especially, the level of TILs has a clinical importance for predicting response to chemotherapy and survival rate in TNBC [72,73]. TNBC patients who have more than 50% of TILs in the tumor showed about 40% of pCR, whereas those with no TILs showed 4% of pCR [74]. In addition, disease-free survival (DFS) and OS were improved in the patients with a high level of TILs [10]. The survival of TNBC patients was increased in a group showing numerous CD8 T-lymphocytes among the TILs compared to those showing few CD8 T-lymphocytes [15,75]. However, a therapeutic strategy to increase the number of TILs is needed to improve survival and therapeutic effect, since only small proportion of TNBCs show a high TIL rate.

Recently, neutrophil–lymphocyte ratio (NLR) has been reported as another prognostic marker predicting clinical outcomes in cancer immunology. Neutrophils have been known to facilitate angiogenesis and disease progression, which increases the possibility of metastasis and recurrence [76,77]. Furthermore, neutrophils inhibit the anti-cancer immunity of T-lymphocytes and natural killer cells [78]. Recent studies showed that the level of NLR in a neoadjuvant chemotherapy setting has prognostic value for pCR in TNBC patients [79,80,81]. Furthermore, a high level of NLR was adversely correlated with DFS and OS [79].

Although there is no clearly defined prognostic marker for predicting the disease progression of TNBC patients, clinical evidence continues to show that the analysis of patients’ immune systems can predict the response to and survival of chemotherapy. Overall, early identification of a patient’s immune environment could be a potential therapeutic strategy for TNBC.

## 5. Available New Drugs

### 5.1. Carboplatin

Carboplatin, a type of platinum salt, induces DNA crosslink strand breaks and results in apoptosis of cells with a dysfunctional repair pathway (Figure 1). Originally, platinum chemotherapeutic agents showed only a modest response in breast cancer except for chemotherapy-naïve tumors [16,82]. Therefore, these drugs were much more available for other cancer types, such as ovarian cancer, than breast cancer. However, the method of classifying TNBC subtype revealed that the BLBC subtype, which accounts for most TNBC tumors, has the characteristics of genomic instability, BRCA1 abnormalities, defected DNA repair pathway, and replication stress. These features are similar to those observed in ovarian cancer [83]. Preclinical and clinical data have accumulated to examine again the effect of platinum agents for TNBC [84].

A multicenter phase II clinical trial evaluated platinum monotherapy to assess biomarkers for patients with metastatic TNBC [85]. Carboplatin and cisplatin showed overall response rates (ORRs) of 18.7% and 32.6%, respectively. Patients harboring *BRCA1/2* mutations (*n* = 11) had a higher ORR (54.5%) than those without *BRCA1/2* mutations (*n* = 66; ORR, 19.7%). In addition, the HRD score was shown to discriminate between responding and non-responding groups to platinum agents in metastatic TNBC.

A phase III Triple-Negative breast cancer Trial (TNT) evaluated the efficacy of either carboplatin or docetaxel in 376 unselected patients with metastatic TNBC [86]. In a randomly distributed population (*n* = 376; 188 carboplatin and 188 docetaxel), there was no evidence that carboplatin has a better response rate than docetaxel (ORR, 31.4% and 34.0%, respectively; *p* = 0.66). However, in patients with germline *BRCA1/2* mutations, the carboplatin-treated group had a higher ORR (68%) than the docetaxel-treated group (ORR, 33%). Interestingly, this responsiveness was not found in those with BRCA1 methylation, low expression of BRCA1 mRNA, or high HRD score tumors [86]. Although the patients with PAM50 basal tumors did not show a better response to carboplatin than docetaxel (ORR, 32.5% vs. 31.0%; *p* = 0.78), those with non-basal-like tumors had a better response to docetaxel than carboplatin (ORR, 72.2% vs. 16.7%; *p* = 0.002). Thus, this study did not reveal molecular evidence to identify who will benefit from treatment with a platinum agent, besides the germline BRCA1/2 mutation status.

Recently, clinical trials of neoadjuvant carboplatin treatment were released in TNBC showing increased pCR in the carboplatin-added arm [87,88]. The phase II trial (NCT01525966) evaluated the efficacy of neoadjuvant carboplatin with combinatory nab-paclitaxel in early stage TNBC, newly diagnosed stage II- III [87]. Sixty-seven patients were enrolled in this study and showed manageable toxicity. The combination of carboplatin and nab-paclitaxel was highly effective in TNBC, and 32 of 67 patients (48%) showed pCR. In addition, higher pCR was associated with “immune-hot” GeparSixto immune signature (GSIS; *p* = 0.005) [89] and DNA repair defect (DRD; *p* = 0.03). However, BRCA status did not show a significant correlation with pCR. GSIS includes 12 immune genes (CCL5, CXCL9, CXCL13, CD80, CD21, CD8A, IGKC, PD-1, CD274 (PD-L1), CTLA4, FOXP3, and IDO1). The result of this study showed that those genes have a potential to be used as clinical markers for the combinatory treatment of carboplatin and nab-paclitaxel. The benefit of carboplatin in early-stage TNBC patients provides a rationale to use platinum-based therapy for (neo)adjuvant treatment, but only currently ongoing randomized phase 3 trials could answer definitely regarding event-free survival (PEARLY trial: NCT02441933; NRG-BR003 trial: NCT02488967)

### 5.2. PARP Inhibitor

PARP is an essential enzyme for DNA repair, cell proliferation, and signaling to other cell-cycle proteins through the mechanism of action of transferring ADP-ribose from NAD+ to target proteins [90,91]. In replicating cells, the inhibition of PARP induces double-strand breaks; therefore, PARP inhibitors have selective toxicity in BRCA1/2-deficient cells with impaired homologous recombination [6].

Olaparib is the first approved PARP inhibitor with an advanced efficacy for patients with germline BRCA-mutated metastatic breast cancer (gBRCAm-BC) (Figure 1 and Table 2). This approval was based on result from OlympiAD [92], an open-label and randomized phase III trial (*n* = 302) evaluating olaparib compared with chemotherapies (capecitabine, vinorelbine, or eribulin). Median progression-free survival was significantly longer in the olaparib group than in the standard therapy group (7.0 months vs. 4.2 months; hazard ratio (HR), 0.58; 95% confidence interval (CI), 0.43-0.80; *p* < 0.001). Patients treated with olaparib monotherapy (*n* = 205) had a higher response rate than those with the standard-therapy (*n* = 97; 59.9% vs. 28.8%).

The EMBRACA study evaluated talazoparib, another PARP inhibitor, for patients with gBRCAm-BC (NCT01945775) [93]. The randomized and open-label trial assigned the patients to receive talazoparib or standard chemotherapies (capecitabine, eribulin, or vinorelbine) in a 2:1 ratio. The talazoparib group showed longer median progression-free survival (PFS) than the standard chemotherapy group (8.6 months vs. 5.6 months; HR = 0.542; *p* < 0.001). The ORR was also higher in the talazoparib group than in the standard chemotherapy group (62.6% vs. 27.2%; *p* < 0.001). The results of this trial showed that talazoparib had a significant benefit compared with the standard chemotherapy for patients with advanced breast cancer and germline *BRCA* mutations, which contributed to the U.S. Food and Drug Administration (FDA)’s approval of talazoparib (Table 2).

Other PARP inhibitors are currently being evaluated in clinical trials: rucaparib (NCT01074970), veliparib (NCT02163694), and niraparib (NCT01905592).

### 5.3. Immumotherapy

Using the immune check point mechanism, tumor cells have the ability to evade recognition and cell death by the host’s adaptive immune system. Thus, this mechanism is considered to be a therapeutic target for effective antitumor immunity. Major checkpoint molecules are programmed cell death ligand 1 (PD-L1) and programmed cell death protein 1 (PD-1). PD-L1 on the surface of tumors bind to PD-1 of cytotoxic T cells (Figure 1), inducing signal transduction to inhibit T cell activation and cause immune tolerance [95]. Although breast cancer has not been known as actively immunogenic tumors, TNBC shows a higher number of tumor-infiltrating lymphocytes, which are regarded as the prognostic marker for antitumor immunotherapies [10], as well as higher PD-L1 expression [20,21] than other breast cancer types.

These results led to clinical trials testing the efficacy of immunotherapies in advanced and metastatic TNBC. Pembrolizumab, an anti-PD-1monoclonal antibody, was evaluated in the phase II KEYNOTE-086 trial in patients with advanced TNBC who had three prior chemotherapies showing an encouraging ORR (23.1%) in PD-L1 positive [22,23]. Subsequently, pembrolizumab was approved based on the results of the KEYNOTE-355 (NCT02819518) double-blind and randomized trial with patients with locally recurrent, unresectable, or metastatic TNBC without previous chemotherapy treatment (Table 2) [24]. Patients who received pembrolizumab plus chemotherapy had significantly longer median PFS than those receiving placebo plus chemotherapy (9.7 months vs. 5.6 months; HR 0.65; 95% CI: 0.49–0.86; one-sided *p* = 0.0012).

Furthermore, the effect of pembrolizumab on patients with early TNBC was investigated in the phase III KEYNOTE-522 trial [25]. Pembrolizumab was used to treat stage II or stage III TNBC patients with neoadjuvant chemotherapy. The rate of patients with a pCR was 64.8% in the pembrolizumab-chemotherapy group and 51.2% in the placebo-chemotherapy group (treatment difference: 13.6%, 95% CI, 5.4 to 21.8; *p* < 0.001). This study showed that neoadjuvant chemotherapy plus pembrolizumab could significantly improve pCR in early TNBC patients, with statistically significant enhanced event-free survival [25]. However, in the KEYNOTE-119 trial, single treatment with pembrolizumab did not show significant improvement compared to chemotherapy in metastatic TNBC patients who failed first-line therapy [26].

Similarly, an anti-PD-L1 monoclonal antibody, atezolizumab, was also approved in combination with protein-bound paclitaxel for patients with unresectable locally advanced or metastatic TNBC without prior chemotherapy based on the Impassion130 (NCT02425891) trial (Table 2) [27]. In patients expressing PD-L1, those who received atezolizumab with protein-bound paclitaxel exhibited significantly longer PFS than those receiving the placebo with protein-bound paclitaxel (7.4 months vs. 4.8 months; *p* = 0.002). The ORR was 53% for the atezolizumab arms, compared with 33% for the placebo arms. Furthermore, the PD-L1-positive population showed better median OS (25.0 months vs. 15.5 months; HR, 0.62; 95% CI, 0.45–0.86).

### 5.4. Antibody-Drug Conjugates

Antibody-drug conjugates (ADCs) are attracting attention as a new anticancer therapy that can payload cytotoxic drugs on monoclonal antibodies through specific linkers. ADCs can specifically deliver high-dose cytotoxic drugs precisely to cancer cells. Recently, novel ADCs have entered clinical studies to evaluate their efficacy for patients with TNBC.

An anti-trophoblast cell surface antigen 2 (Trop2) ADC, Trodelvy (sacituzumab govitecan), was FDA approved for the treatment of TNBC patients who have received at least two prior therapies (Table 2). Trop2 has recently shown potential as a therapeutic target due to its specific overexpression in cancer cells [96]. It is also involved in embryonic development and oncogenic signaling pathways, such as the MAPK-ERK1/2 pathway [97,98]. Trodelvy is the anti-Trop2 humanized monoclonal antibody linked to SN-38, an irinotecan metabolite known as a topoisomerase I inhibitor, through a maleimide–polyethylene glycol–acid-sensitive cleavable (carbonate) linker [34,99]. Following accelerated approval based on results of a phase II clinical trial, the randomized phase III ASCENT (NCT01631552) trial showed longer PFS with sacituzumab govitecan than with standard chemotherapy, leading to approval of the new ADC for refractory metastatic TNBC [94]. Patients treated with sacituzumab govitecan (*n* = 108) showed a PFS of 5.5 months, whereas those with standard chemotherapy had a PFS of 1.7 months (HR, 0.41; *p* < 0.0001).

Furthermore, there is another promising ADC, trastuzumab deruxtecan (T-DXd, formerly DS-8201a). T-DXd is a HER2-targeted ADC payloaded with topoisomerase I inhibitor, an exatecan derivative [35]. The phase I study of T-DXd was conducted in TNBC patients with HER2-low tumors (defined as IHC 1+/ISH negative or 2+/ISH negative), as well as strong HER2-positive breast cancers [35]. HER2-low breast cancer patients (*n* = 54) showed 37% of the independent-central-review-confirmed ORR and 44.4% of the investigator-reported-confirmed ORR, with a median response duration of 10.4 months. This trial indicated that patients with HER2-low breast cancer could benefit from this novel HER2-targeted ADC, and currently, a phase 3 trial is ongoing.

## 6. Upcoming and Potential Targeted Therapies

### 6.1. AKT Inhibitor

Aberrant activation of the PI3K/AKT pathway is frequently shown in TNBC patients (Figure 1), which suggests that AKT inhibitors might be effective for those patients (Table 3). The randomized phase II clinical trial of ipatasertib, an AKT inhibitor, with paclitaxel was evaluated in TNBC patients as first-line treatment (NCT02162719; LOTUS) [29]. Patients treated with paclitaxel/ipatasertib (*n* = 62) showed longer median PFS than those with paclitaxel/placebo (*n* = 62; 6.2 months vs. 4.9 months; *p* = 0.037). Especially in the group of PIK3CA, AKT1, and PTEN-altered patients (n = 42), ipatasertib had better PFS than those treated with placebo (9.0 months vs 4.9 months). The ongoing phase II and III clinical trials are evaluating the efficacy of ipatasertib in combination with paclitaxel for PIK3CA/AKT1/PTEN-altered TNBC (NCT03337724; IPATunity130).

In addition, a phase III double-blind randomized clinical study is currently enrolling patients with locally advanced or metastatic TNBC to investigate the benefit of capivasertib, another AKT inhibitor, with paclitaxel as first-line treatment (NCT03997123; CAPItello-290).

### 6.2. Anti-Androgen Targeted Therapy

A phase II clinical trial of bicalutamide, an androgen-blocking agent, was reported in patients with metastatic AR-positive TNBC (Table 3) [100]. Patients who showed more than 10% AR nuclear staining were considered to be AR positive. Among patients with AR-positive tumors, single treatment with bicalutamide showed a clinical benefits rate of 19%, including complete or partial response, or stable disease more than 6 months. The median PFS was 12 weeks (range, 11–22 weeks). Single treatment with bicalutamide was well tolerated without high-grade adverse events. This study demonstrated the proof of concept for AR-targeted therapy in AR-positive TNBC.

Another phase II clinical trial with enzalutamide (Table 3), a new generation AR inhibitor, was reported in patients with advanced-stage AR-positive TNBC, and 55% of them had high AR expression (immunohistochemistry ≥ 10%) [101]. At 16 weeks, 35% of patients showed clinical benefits for enzalutamide.

There are various ongoing clinical trials for AR-targeted therapies with paclitaxel (NCT02689427) or palbociclib (NCT02605486) in metastatic TNBC (Table 3). Moreover, in a preclinical study on LAR-subtype TNBC, a relationship was shown between the PI3K/AKT pathway and AR-targeted therapy. A phase Ib/II clinical trial is ongoing to evaluate the combination of taselisib, a PI3K inhibitor, and enzalutamide in metastatic TNBC (Table 3; NCT02457910).

### 6.3. Other Therapies

Aberrant MEK activation and an abnormality of the RAS/MAPK pathway in TNBC and BLBC led to the combination of a MEK inhibitor and chemotherapies or other targeted therapies (Figure 1). A phase I clinical trial with gemcitabine and trametinib (an inhibitor of MEK1/2) was studied in patients with several solid tumors, including metastatic TNBC [102]. The overall response was not significant; however, the only pCR was shown in patients with metastatic TNBC. A phase II trial of single-agent trametinib followed by trametinib in combination with GSK2141795 (AKT inhibitor) is ongoing in patients with advanced TNBC (NCT01964924). There are many other approved clinical trials for metastatic TNBC. A histone deacetylase inhibitor is under evaluation as monotherapy (NCT02623751). In addition, a phase II clinical trial with a c-MET inhibitor (cabozantinib) was conducted for metastatic TNBC (NCT01738438).

## 7. Conclusions

The biggest obstacle in terms of drug development would be the heterogeneity of TNBC, leading to dissecting this 15% of whole breast cancers into small target populations to be studied in clinical trials. Nevertheless, successful targets have been found, including BRCA1/2, PD-L1, and Trop2, leading to the approval of PARP inhibitors, immune checkpoint inhibitors, and sacituzumab govitecan in recent years. HER2, AKT, and AR are also under investigation for use in the clinic. These struggles have yielded potential new treatments for this dismal and hard to treat subtype of breast cancers. More biologic dissection and subsequent tailored clinical trials are warranted for future success.

## Figures and Tables

**Figure 1 pharmaceuticals-14-01008-f001:**
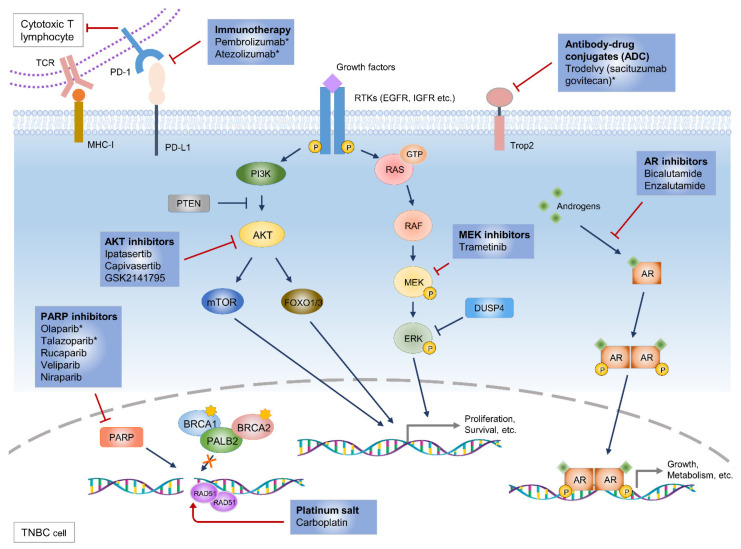
Scheme of potential molecular targets and therapeutic agents including FDA-approved and upcoming targeted therapies in TNBC. Extracellular stimuli, such as growth factors, initiate PI3K/AKT and RAS/MAPK signaling pathways to activate transcription of target genes involved in proliferation, survival, and tumorigenesis. There are potential targeted therapeutics that inhibit these pathways, including AKT inhibitors or MEK inhibitors. In the case of TNBC tumors harboring BRCA1/2 mutations, PARP replaces them, allowing DNA damage to repair and tumor cells to survive. Thus, PARP inhibitors are commonly prescribed for BRCA1/2 mutated TNBC patients. Carboplatin, a platinum salt, induces DNA crosslink strand breaks and causes the apoptosis of tumor cells with a dysfunctional repair pathway. Androgen receptor (AR) is activated by binding of androgen, triggering dimerization and translocation of androgen-AR complex to stimulate cell growth and metabolism in TNBC. AR inhibitors compete with androgens to bind AR and block the AR signaling pathway. Cytotoxic T-lymphocytes recognize the neo-antigen of tumor cells binding to major histocompatibility complex (MHC)-I and kill the tumor cells. To avoid anti-cancer immunity of cytotoxic T-lymphocytes, TNBC tumor cells exhibit anti-programmed death ligand 1 (PD-L1), which binds to PD-1 of the T-lymphocyte. Immunotherapeutic antibodies have been introduced to suppress immune evasion in TNBC tumors. Recently, antibody-drug conjugates (ADC) have been approved by the FDA for clinical use in TNBC. Blue arrow: signaling transduction, red arrow: action of blockers and inhibitors, yellow star: mutations, yellow circle: phosphorylation, orange circle: neo-antigen peptide, *: U.S. Food and Drug Administration (FDA)-approved drugs.

**Table 1 pharmaceuticals-14-01008-t001:** Characteristics and potential therapies of TNBC subtype.

TNBC Subtype	Histology	Characteristics	Molecular and Pathway Alterations	Potential Therapeutics	Reference
Basal-like 1	Ductal carcinoma and invasive ductal carcinoma tumors	Cell-cycle-regulating and DNA repair pathway	MYC, PIK3CA, KRAS, FGFR1, AKT2, BRCA1, TP53, and RB1 amplifications	Platinum salts (carboplatin, cisplatin, etc.) PARP inhibitors (olaparib, talazoparib, etc.)	[4,5,9,10,11,12,13,14,15,16]
Basal-like 2	Ductal carcinoma and invasive ductal carcinoma tumors	Cell-growth-related pathway	EGFR, MET, Wnt/b-catenin, mTOR, and IGF1R pathways	Platinum salts (carboplatin, cisplatin, etc.) PARP inhibitors (olaparib, talazoparib, etc.) Growth factor inhibitors (lapatinib, gefitinib, and cetuximab)	[9,10,17,18,19]
Immunomodulatory	Medullary tumors	Immune-cell-associated signaling pathway	CTLA4, NK cell, Th1/Th2, NF-kB, TNF, T cell signaling, dendritic cell, and BCR pathways	Immune checkpoint inhibitors (pembrolizumab, atezolizumab, and others)	[10,20,21,22,23,24,25,26,27]
Mesenchymal like	Sarcoma-like and squamous epithelial cell-like tumors	Cell migration- and ECM-related pathway	Wnt, TGFb, and ECM pathways	Tyrosine kinase inhibitors, mTOR inhibitor, EMT inhibitor (eribulin mesylate)	[9,28]
Mesenchymal stem-like	Sarcoma-like and squamous epithelial cell-like tumors	Stem-cell-related pathway	BMP2, ENG, ITGAV, NGFR, PDGFR, THY1, KDR, and VCAM1	PI3K inhibitor, Src antagonists, antiangiogenic drugs	[9,29]
Luminal androgen receptor	Apocrine tumors	Hormone-related pathway	Androgen and estrogen metabolism, steroid biosynthesis, tyrosine metabolism, and ATP synthesis	Anti-androgen therapies (bicalutamide, enzalutamide, etc.)	[30,31,32,33,34,35]

**Table 2 pharmaceuticals-14-01008-t002:** Drugs approved by FDA for patients with TNBC.

Drug Name	Target	Dosage Form	FDA Approved Date	Clinical Study Number	Study
Olaparib	PARP	Chemical	January, 2018	NCT02000622	[92]
Talazoparib	PARP	Chemical	October, 2018	NCT01945775	[93]
Pembrolizumab	PD-1	Monoclonal antibody	November, 2020	NCT02819518	[24]
Atezolizumab	PD-L1	Monoclonal antibody	March, 2019	NCT02425891	[27]
Trodelvy (sacituzumab govitecan)	Trop2, Topoisomerase I	ADC	April, 2020	NCT01631552	[94]

ADC: antibody-drug conjugates; PARP: poly (ADP-ribose) polymerase; PD-1: programmed cell death protein 1; PD-L1: programmed cell death ligand 1; Trop2: trophoblast cell surface antigen 2.

**Table 3 pharmaceuticals-14-01008-t003:** Ongoing clinical trials of upcoming targeted therapies for patients with TNBC.

Targets	Disease Setting	Breast Cancer Subtype	Phase	Therapies (Alone or Combination)	Control Treatment	Clinical Trial Reference Number
AKT	Metastatic	TNBC	II	Ipatasertib (GDC-0068) + paclitaxel	Paclitaxel	NCT02162719
	Locally advanced or metastatic	TNBC or hormone receptor-positive, HER2-negative BC	III	Ipatasertib + paclitaxel	Paclitaxel	NCT03337724
	Locally advanced (inoperable) or metastatic	TNBC	III	Capivasertib (AZD5363) + paclitaxel	Paclitaxel	NCT03997123
Androgen	Neoadjuvant	Androgen receptor-positive TNBC	IIb	Enzalutamide + paclitaxel	Paclitaxel	NCT02689427
	Metastatic	Triple negative, androgen receptor positive BC	I/II	Bicalutamide + palbociclib	NA	NCT02605486
	Metastatic	Androgen receptor-positive TNBC	Ib/II	Enzalutamide + taselisib (GDC-0032)	NA	NCT02457910
MEK	Metastatic	TNBC	II	Trametinib + GSK2141795	NA	NCT01964924
HDAC	Advanced or recurrent	All	I	KHK2375 + exemestane	NA	NCT02623751
c-MET	Metastatic	TNBC	II	Cabozantinib	NA	NCT01738438

BC: breast cancer; HER2: human epidermal growth factor receptor 2; HDAC: histone deacetylases; NA: not applicable; TNBC: triple-negative breast cancer.

## Data Availability

Data is contained within the article.

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
