# Peer review of "Molecular Targets and Promising Therapeutics of Triple-Negative Breast Cancer"

_pharmaceuticals, 2021, doi:10.3390/ph14101008_

Round 1

Reviewer 1 Report

The authors provide a comprehensive overview of triple negative breast cancer and its treatment options. The review covers all important areas and worth of publication. Some minor comments are as follows:

Line 349-357: the paragraph described T-DxD as her 2 targeted top inhibitor. The reference also does not claim the patients as TNBC, but Her2+ low. The authors might want to clarify relevance for TNBC.

Table 3: header is not properly formatted

Table1 might benefit from a column showing the relevant references.

Reviewer 2 Report

This paper by Ryu et al. entitled «Molecular targets and promising therapeutics of triple-negative breast cancer» is an interesting review about the novel therapeutic strategies for triple-negative breast cancer. Some points however remain unclear.

Introduction:

  • The authors said: “its mainstream treatment is endocrine 21 therapy with or without targeted agents to interrupt estrogen-signaling pathways and 22 other critical pathways for cancer survival”. What are the targeted agents and the other critical pathways mentioned?
  • The introduction should be more developed. For example, what is the definition of personalized therapy and the problematic with the heterogeneity of triple-negative breast cancer.

Figure 1:

  • Where is Figure 1 cited in the text?
  • There are “*” cited in Figure 1 but they are not described in the legend. Please explain.
  • It should be relevant to add a legend for the arrows as there are different colors (blue and red).
  • It is not clear how anti-PDL-1 works. It works with the immune system, but it is not well represented in Figure 1.

Section Carboplatin:

  • Carboplatin is a chemotherapy drug. The authors should have discussed about the other chemotherapy agents to treat triple-negative breast cancer.

Table 2: The authors should add a legend with the abbreviations cited in the table. Please correct “Olaparib”.

Table 3:

  • The column “Control are treatment” does not make sense. Please revise.
  • The column “Breast cancer” and “Phase” should be reversed for a better understanding.
  • The authors should add a legend with the abbreviations cited in the table.

Conclusion: the conclusion could be more developed and be more focused on the challenge raised by the heterogeneity of triple-negative breast cancer due to the different subgroups.

The authors should add an abbreviation list.

There are some typos and grammatical mistakes through the text. The manuscript should be proofread before publication. English must be revised. The manuscript should be read by an English-speaking person.

Page 9, line 22: …who receiving the placebo…(either who receive or receiving).

Page 9, line 31: …by the FDA for treatment… Please rephrase

Author Response

This manuscript is a resubmission of an earlier submission. The following is a list of the peer review reports and author responses from that submission.